# Development and Application of Droplet Digital PCR Assay for the Detection of Watermelon Silver Mottle Virus and Melon Yellow Spot Virus

Huijie Wu [1,2], Mei Liu [1], Wenyang Li [1], Min Wang [3], Junqing Xiu [1], Bin Peng [1], Yanping Hu [3], Baoshan Kang [1,2], Liming Liu [1] and Qinsheng Gu [1,*]

[1] Zhengzhou Fruit Research Institute, Chinese Academy of Agricultural Sciences, Zhengzhou 450009, China; wuhuijie@caas.cn (H.W.)

[2] Zhongyuan Research Center, Chinese Academy of Agricultural Sciences, Xinxiang 453500, China

[3] Key Laboratory of Vegetable Biology of Hainan Province, The Institute of Vegetables, Hainan Academy of Agricultural Sciences, Haikou 571199, China

[*] Correspondence: guqinsheng@caas.cn; Tel.: +86-371-65330997

**Abstract:** Watermelon silver mottle virus (WSMoV) and melon yellow spot virus (MYSV) (Tospoviridae, *Orthotospovirus*) are responsible for silver mottle mosaic and yellow spot symptoms, posing threats to melon (*Cucumis melo*), watermelon (*Citrullus lanatus*), and cucumber and leading to significant economic losses in China. Early disease detection and monitoring of these two viruses are necessary for disease management, for which a rapid, reliable, and adaptable diagnostic method is required. In this study, using a droplet digital PCR (ddPCR) method, the conserved region of the nucleocapsid gene (*N* gene) sequence was detected in WSMoV and MYSV. The probes and primers for WSMoV and MYSV did not detect other relevant cucurbit viruses, and the specificity reached 100%. Although both qPCR and ddPCR exhibited good reproducibility, the reproducibility of ddPCR was better than that of qPCR. The reproducibility of ddPCR was proved to be 100%. Moreover, ddPCR exhibited a good linear correlation with varying concentrations of targets. The detection limits of WSMoV and MYSV in ddPCR were 18 and 9 copies/μL and were approximately 12- and 18-times more than those in qPCR, respectively. Finally, 62 samples collected from the field (including infected melon, watermelon, and weeds) were further evaluated for the presence of WSMoV and MYSV. The field samples exhibited 91.94% and 51.61% positivity rates in ddPCR assays for WSMoV and MYSV, respectively; the rates were higher than those in qPCR (59.68% and 43.39%, respectively). The results indicated that ddPCR has a higher accuracy than qPCR. Therefore, ddPCR could be used in the clinical diagnosis of early infections of WSMoV and MYSV. To the best of our knowledge, this is the first study to establish a ddPCR method for the detection of WSMoV and MYSV. The application of this method for differential detection of MYSV and WSMoV will help in understanding the epidemics caused by these two important viruses and provide important information for the early detection, monitoring, and rapid extermination of infection.

**Keywords:** watermelon silver mottle virus (WSMoV); melon yellow spot virus (MYSV); droplet digital PCR; detection methods

## 1. Introduction

Watermelon and melon are popular fruits, providing rich nutrients. However, their production faces serious threats due to diseases and pests throughout the growing season. Viral diseases, such as melon yellow spot orthotospovirus (MYSV), watermelon silver mottle virus (WSMoV), cucurbit chlorotic yellows virus (CCYV), zucchini yellow mosaic virus (ZYMV), tomato leaf curl New Delhi virus (ToLCNDV), cucumber mosaic virus (CMV), and cucumber green mottle mosaic virus (CGMMV), cause great losses in cucurbit

production. MYSV mainly damages wax gourds, watermelons, netted melons, and cucumbers [1]. In cucumber, MYSV causes severe productivity and quality losses [2]. Our field disease survey demonstrated that in melon, severe yellow spot occurred on the leaves, and patches of spots seriously affected plant photosynthesis and caused leaf death. WSMoV was first reported in Japan [3]. Subsequently, it was reported in Taiwan [4], India [5], and Thailand [6]. In China, WSMoV was reported in watermelon [7]. WSMoV causes significant losses in Cucurbitaceae plants. Watermelons infected by WSMoV exhibit leaf deformities, with silver-gray patches appearing on the leaf edges. In severe cases, the entire leaf becomes silver gray; the surface of the fruit also appears faint silver gray. When the fruit is deformed, the flesh color is not uniform. This phenomenon causes uneven fruit coloring and reduces economic value. In China, the virus is gradually spreading to the watermelon growing areas in the north. Both WSMoV and MYSV are from the family Tospoviridae and genus *Orthotospovirus* and are important viruses. Both of them usually coinfect crops, such as *Siraitia grosvenorii*, *Chieh-Qua* [8], and cucumber [9]. Both MYSV and WSMoV are transmitted by *Thrips palmi* Karny (Thysanoptera: Thripidae), which is a persistent leaf feeder [1]. The control and prevention of MYSV and WSMoV are very difficult because both viruses infect not only cucurbits but also weeds. Therefore, to prevent the spread of the virus and large-scale outbreaks that harm agricultural production, research and improvements in virus detection technology should be promoted; epidemic monitoring of the virus should be conducted, and an early warning and monitoring system should be established. An efficient and accurate diagnostic method is very urgently needed for the detection of WSMoV and MYSV in nurseries and on fields.

Currently, many detection methods are used to identify WSMoV or MYSV. They include electron microscopy [4], reverse-transcriptase polymerase chain reaction (RT-PCR) [10], enzyme-linked immunosorbent assay (ELISA), Western blotting [11], and double-antibody sandwich ELISA (DAS-ELISA) [12]. However, these methods have certain limitations [13]. Recently, SYBR Green I real-time PCR was used to identify WSMoV [14]. Moreover, high-throughput sequencing (HTS) [8,15] and Oxford nanopore sequencing technology [9] were used to identify and analyze these viruses.

Droplet digital PCR (ddPCR) (or RT-dPCR; digital PCR) is an emerging and innovative third-generation PCR technology for the absolute quantification of nucleic acids without the requirement of a standard curve. It is based on the number of positive and negative partitions using Poisson statistics after the reaction is terminated [16]. It is a highly sensitive, specific, and accurate method [17,18], particularly when the amount of nucleic acid is relatively low. For ddPCR, specific devices and additional costs of TaqMan probes are needed in each reaction, compared with simple RT-PCR. Currently, ddPCR is applied in medical and clinical studies, and its use is increasing in the field of plant pathology. For example, it was used for the detection of pathogenic bacteria, such as *Erwinia amylovora*, *Ralstonia solanacearum* [19], and *Xylella fastidiosa* [20]; fungi [21]; and plant viruses [22]. The ddPCR method has been widely used for the detection of nucleic acids with low abundance and diagnosis of infectious diseases [23]. In addition, it is used to detect viruses in cucurbits. qPCR and ddPCR assays have been developed for both WMV and ZYMV [24]. The RT-dPCR method is highly reliable and useful for the detection and quantification of CGMMV [25]. However, currently, no ddPCR assays are reported for the molecular detection of both MYSV and WSMoV and early disease diagnosis.

In this study, we applied the ddPCR method to simultaneously detect the two viruses, which may provide another important tool for epidemiological surveillance. The ddPCR assay was established for the detection of MYSV and WSMoV targeting the *N* gene in infected plants. The specificity, linearity, reproducibility, and sensitivity of the ddPCR assay were estimated. Both RT-qPCR and RT-dPCR assays were performed to detect the natural infection of MYSV and WSMoV in cucurbits and weed samples. The RT-dPCR detection method for MYSV and WSMoV may provide effective technical means for accurate quantification of their infection in nursery or field samples. This study provided technical support for monitoring the occurrence of these viruses at an early stage.



## 2. Materials and Methods

### 2.1. Sample Preparation

All samples were collected from the fields four times and from four provinces in China: in Hainan in April 2022, in Yunnan in June 2021, and in Shandong and Zhejiang in July 2023. The samples were from the field or greenhouse and stored at −80 °C. In addition, leaves infected with CGMMV, CCYV, ToLCNDV, and CMV were identified, collected, and stored in our laboratory.

Total RNA was extracted from infected samples using RNA Extraction Kit (Tiangen, Beijing, China) as per the manufacturer's instructions. RNA concentration was determined using NanoDrop 1000 spectrophotometer (Thermo Fisher, Waltham, MA, USA).

RNA was reverse transcribed with a reaction mixture (20 μL) containing 4 μL of 5× MonScript™ RTIII Super Mix (Monad, Shanghai, China), 2 μg total RNA (the volumes were added according to the concentrations of RNA), and ddH$_2$O to make up the volume to 20 μL. The protocol was as follows: 25 °C for 10 min, 50 °C for 15 min, and 85 °C for 5 min. The products were stored at −80 °C.

### 2.2. Construction of Standard Plasmids

*N* genes of WSMoV and MYSV was amplified (primers WSMoV-S-F: 5′-GCTGTTCCAG-GGTTACTTTC-3′/WSMoV-S-R: 5′-GGACTCCACTCCCGGATTTA-3′ and MYSV-cp-F: 5′-TTAAACTTCAATGGACTTAGATC-3′/MYSV-cp-R: 5′-ATTCAACATCAGCAAGTCAA-3′), which yielded 609- and 886-bp amplicons, respectively. The amplicons were cloned into pTOPO-Blunt vector from Aidlab Biotechnologies Co., Ltd. (Aidlab, Beijing, China). The pTOPO-WSMoV and pTOPO-MYSV plasmids were purified and quantified. The number of plasmid DNA copies (copies/μL) was calculated as [plasmid concentration (ng/μL) × $10^{-9}$]/[the length of the plasmid (bp) × 660 × (6.02 × $10^{23}$)] [26].

### 2.3. Primer and TaqMan Probe Design

The primers and probes for WSMoV and MYSV isolates were designed based on the complete segment S sequences of WSMoV (GenBank Accession Number: NC_003843.1) and MYSV (GenBank Accession Number: KX711613.1) for ddPCR analyses. Two pairs of primers of WSMoV and MYSV were WSMoV-ddF and WSMoV-ddR (5′-ACTTCTCCACCT-ACGAGCAA-3′ and 5′-AGCCTGTTTGAACATAACATCCA-3′) and MYSV-ddF and MYSV-ddR (5′-TCACCTCATCTCTCACATCCG-3′ and 5′-ACACCATGTCTACCGTTGCT-3′), respectively. The probe sequences were WSMoV-ddprob-FAM-5′-TGTGAAGCAGACAGAA-CCTTAGCCACT-3′-BHQ1 and MYSV-ddprob-VIC-5′-TCCGACTTGCCACCACTGAGAA-GT-3′-BHQ1. All sequences were synthesized by Sangon Biotech Co., Ltd. (Sangon, Shanghai, China).

### 2.4. TaqMan-Based RT-qPCR

The primers and probes of WSMoV and MYSV were the same for ddPCR and RT-qPCR. qPCR detection of WSMoV and MYSV was conducted on LightCycler 480 real-time PCR system (Roche, Basel, Switzerland). The 30 μL reaction mix included the following: 15 μL of 2× MonAmp™ Taqman qPCR Mix (None ROX) (Monad), 1.8 μL of each primer (10 μM), 1.0 μL (10 μM) of WSMoV probe, 0.75 μL (10 μM) of MYSV probe, 5 μL of template, and ddH$_2$O to make up the volume to 30 μL. The protocol was 95 °C for 4 min, followed by 40 cycles of 95 °C for 15 s and 55 °C for 30 s.

### 2.5. RT-dPCR

A TD-1 Droplet Digital PCR system (TargetingOne, Beijing, China) was used to perform RT-dPCR. The 30 μL reaction mix contained 7.5 μL 4× ddPCR™ Supermix mixture (TargetingOne), 4.48 μL (600:330 nM and 600:250 nM, respectively) of the primer and probe for WSMoV and MYSV, 5 μL template, and ddH$_2$O to make up the volume to 30 μL. Further, 160 μL micro drip oil and 30 μL ddPCR mixture were transferred to the chip to generate

droplets, which were subjected to PCR protocol as follows: 95 °C for 4 min, 40 cycles of 94 °C for 15 s and 55 °C for 30 s, and storage at 4 °C.

To determine the optimal annealing temperature for RT-dPCR, a temperature range of 54, 55, 56, 57, 58, 59, 60, and 61 °C was set. The primer-to-probe concentrations of WSMoV (600:330, 660:400, 660:300, and 800:250 nm) and MYSV (600:250, 660:330, 660:300, and 800:250 nm) were optimized. Each ddPCR assay was performed three times.

### 2.6. Assessment of the Specificity of WSMoV and MYSV Primers and Probes

To evaluate the specificity of the WSMoV and MYSV primers and probes, the samples infected with CGMMV, CCYV, ToLCNDV, and CMV were subjected to RT-dPCR. The negative control was DEPC-treated water without template (NTC). The intra-assay variability was assessed using three independent assays.

### 2.7. Assessment of Linearity, Reproducibility, and Detection Limits

The repeatability of RT-qPCR and RT-dPCR was evaluated by analysis with the same volume and run system of pTOPO-WSMoV and pTOPO-MYSV plasmids. Five concentrations of WSMoV plasmid log (copies/µL) were 4.18, 3.65, 3.20, 2.74, and 2.30. Five concentrations of WYSV plasmid log (copies/µL) were 3.81, 3.32, 2.93, 2.68, and 2.54. Each sample had three replicates. Coefficients of variation (CVs) were measured based on the standard means of three replicates.

To assess the linearity, a series of dilutions of WSMoV ($3.0 \times 10^5$, $3.0 \times 10^4$, $3.0 \times 10^3$, $1.5 \times 10^3$, $0.5 \times 10^3$, $0.1 \times 10^3$, and $0.02 \times 10^3$ copies·µL$^{-1}$) and MYSV ($2.0 \times 10^5$, $2.0 \times 10^4$, $2.0 \times 10^3$, $1.0 \times 10^3$, $0.3 \times 10^3$, $0.06 \times 10^3$, and $0.03 \times 10^3$ copies·µL$^{-1}$) was used. The copy numbers were determined based on the concentrations and the length of plasmids. Each sample was tested three times.

To analyze the detection limits of qPCR and ddPCR, the pTOPO-WSMoV and pTOPO-MYSV plasmids were diluted according to the plasmid copy numbers, which were, in turn, determined based on the length of the plasmid mentioned above. Serial dilutions according to 500, 300, 200, 100, 50, 20, 10, 5, 2, and 1 copies·µL$^{-1}$ and DEPC-treated water without template were used to detect the detection limits, respectively.

### 2.8. Assessment of Naturally Infected Samples

To assess the clinical effects, naturally infected melon, watermelon, and weed samples in field were collected for RT-PCR and RT-dPCR as per the optimized protocols described above. In total, 62 samples were collected from different fields in Hainan, Shandong, and Zhejiang Provinces between 2021 and 2023. All samples were analyzed using real-time RT-PCR and RT-dPCR for both WSMoV and MYSV at the same time. The positive detection rate of the two methods was evaluated. Negative and positive control reactions were performed at the same time.

### 2.9. Data Analysis

All data in this study were presented as the mean ± SD. GraphPad Prism 5.0 and Excel2016 Software were used to analyze data. The CV values were calculated according to the methods described previously [27].

### 3. Results

#### 3.1. Optimization of the Annealing Temperature of WSMoV and MYSV

To define the best annealing temperature, the temperature range of 54 to 61 °C was analyzed. For MYSV, a temperature of 56–61 °C could not improve fluorescence amplitude separation and slightly reduced specificity between the fluorescence amplitudes of the negative and positive droplets; 54 and 55 °C were the optimal annealing temperatures (Figure 1A). For WSMoV, a temperature of 56–61 °C resulted in no difference between the negative and positive droplets; a temperature of 54 °C reduced the specificity. Therefore, 55 °C was the optimal annealing temperature for WSMoV (Figure 1B). In addition, combin-

ing the optimal annealing temperature of MYSV and WSMoV, 55 °C was considered the optimal annealing temperature in this study. Consequently, the primer/probe sets were observed to specifically detect the target virus, and the optimal annealing temperature of 55 °C was used for further ddPCR and qPCR experiments.

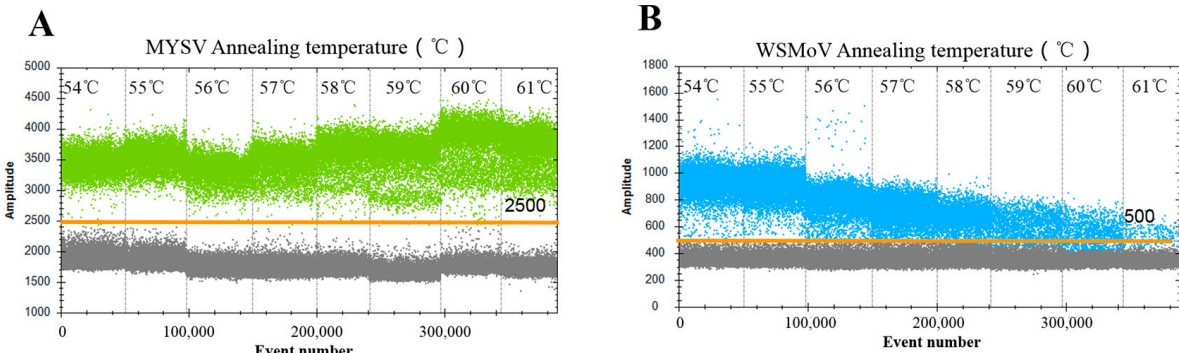

**Figure 1.** Optimization of the annealing temperature for the ddPCR assay. To determine the optimal annealing temperature, annealing temperatures of 54, 55, 56, 57, 58, 59, 60, and 61 °C were assessed. Light green dots represent the positive MYSV (**A**); blue dots represent the positive WSMoV (**B**); the orange line indicates the threshold; the droplets above the lines are positives (blue and light green dots), and the droplets below lines are positives and negatives (gray dots). The X and Y coordinates indicate the number of droplets and fluorescence signal intensity, respectively.

### 3.2. Optimization of Primer and Probe Concentration Ratio for WSMoV and MYSV

To further construct the detection system for MYSV and WSMoV, the concentration of primers and probes was optimized. When the concentration ratios of MYSV and WSMoV primers and probes were 600:250 (Figure 2A) and 600:330 nM (Figure 2B), respectively, in the same system, the fluorescence amplitudes between positive and negative droplets indicated the optimal separation. Therefore, the optimal temperature and concentration of primers and probes were 55 °C and 600:250 nM for MYSV and 600:330 nM for WSMoV, respectively, for the subsequent ddPCR assay.

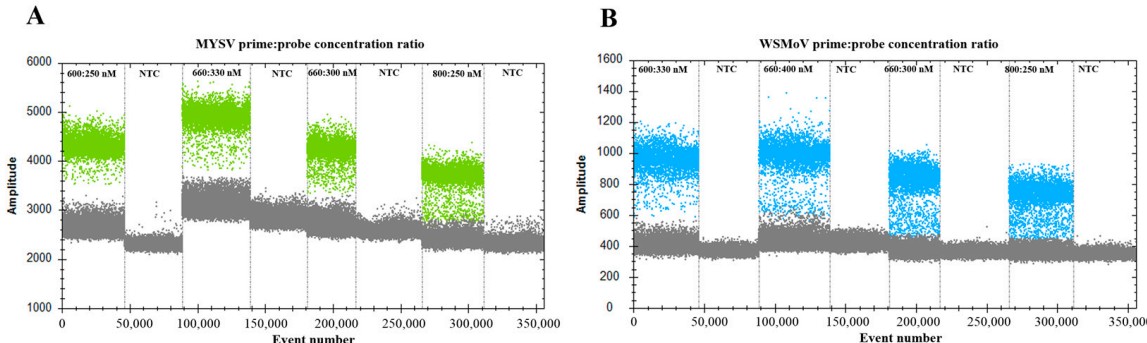

**Figure 2.** Optimization of primer and probe concentration ratio for WSMoV and MYSV. (**A**) Optimization of the primer-to-probe concentration ratio for MYSV (600:250, 660:330, 660:300, and 800:250 nm) and (**B**) WSMoV (600:330, 660:400, 660:300, and 800:250 nm). NTC, no-template control.

### 3.3. Evaluation of the Specificity of Detection of MYSV and WSMoV

To assess the specificity of the ddPCR assay, the aforementioned optimized conditions were used. Six melon and watermelon viruses, including MYSV, WSMoV, CGMMV, ZYMV, ToLCNDV, and WMV, were used as templates, and the assay was performed with primers and probes specific for MYSV and WSMoV. The results revealed that in the negative template control, no other virus was detected; the probes and primers targeting MYSV and WSMoV could not detect other viruses except MYSV and WSMoV, respectively (Figure 3). The specificity evaluation for both was 100%.

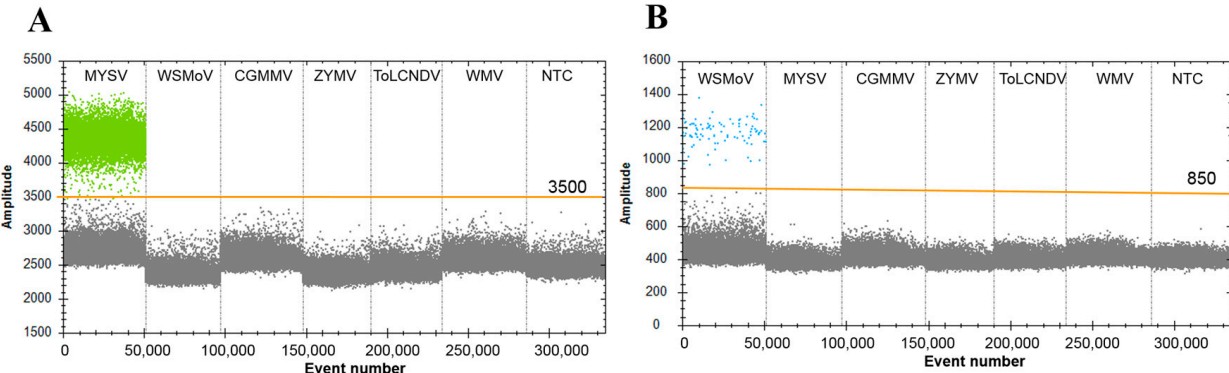

**Figure 3.** Evaluation of the specificity of detection of MYSV and WSMoV in ddPCR assay. (**A**) The specificity of detection of MYSV and (**B**) the specificity of detection of WSMoV. Each sample is divided by orange lines. The fluorescence amplitudes of watermelon silver mottle virus (WSMoV), melon yellow spot virus (MYSV), cucumber green mottle mosaic virus (CGMMV), zucchini yellow mosaic virus (ZYMV), tomato leaf curl New Delhi virus (ToLCNDV), watermelon mosaic virus (WMV), and the no-template control (NTC) are indicated.

*3.4. Assessment of Reproducibility of qPCR and ddPCR*

A series of dilutions of pTOPO-WSMoV and pTOPO-MYSV plasmids was used to analyze the reproducibility. The inter-assay CV of WSMoV was 0.01–0.74% as per ddPCR and 0.54–2.64% as per qPCR. The inter-assay CV of MYSV was 0.14–0.38% as per ddPCR and 0.18–0.98% as per qPCR. The results revealed that, although both qPCR and ddPCR exhibited good reproducibility, the reproducibility of ddPCR was better than that of qPCR. The reproducibility of ddPCR was 100% (Table 1).

**Table 1.** Reproducibility of the ddPCR and qPCR assay.

| Name | Concentration Plasmid Log (copies/µL) | ddPCR Inter-Assay Variation (Reproducibility) | | | | | qPCR Inter-Assay Variation (Reproducibility) | | | | |
|---|---|---|---|---|---|---|---|---|---|---|---|
| | | Rep.1 Log (copies/µL) | Rep.2 Log (copies/µL) | Rep.3 Log (copies/µL) | SD | CV | Rep.1 (Ct) | Rep.2 (Ct) | Rep.3 (Ct) | SD | CV |
| WSMoV | 4.18 | 4.17 | 4.17 | 4.17 | 0.00 | 0.01% | 24.9 | 25.2 | 25.4 | 0.27 | 1.08% |
| | 3.65 | 3.64 | 3.64 | 3.64 | 0.00 | 0.06% | 27.1 | 26.6 | 26.8 | 0.22 | 0.83% |
| | 3.20 | 3.20 | 3.20 | 3.20 | 0.00 | 0.07% | 28.2 | 28.5 | 28.2 | 0.15 | 0.54% |
| | 2.74 | 2.76 | 2.75 | 2.76 | 0.01 | 0.29% | 29.7 | 29.3 | 29.4 | 0.22 | 0.76% |
| | 2.30 | 2.30 | 2.32 | 2.28 | 0.02 | 0.74% | 30.8 | 31.5 | 32.5 | 0.84 | 2.64% |
| MYSV | 3.81 | 3.81 | 3.82 | 3.82 | 0.01 | 0.14% | 23.0 | 23.4 | 22.9 | 0.23 | 0.98% |
| | 3.32 | 3.33 | 3.32 | 3.35 | 0.01 | 0.37% | 24.9 | 25.0 | 24.9 | 0.05 | 0.18% |
| | 2.93 | 2.93 | 2.92 | 2.92 | 0.01 | 0.18% | 26.4 | 26.6 | 26.3 | 0.15 | 0.57% |
| | 2.68 | 2.67 | 2.67 | 2.67 | 0.00 | 0.14% | 27.7 | 27.5 | 27.4 | 0.18 | 0.65% |
| | 2.54 | 2.52 | 2.53 | 2.52 | 0.01 | 0.38% | 28.4 | 28.6 | 28.8 | 0.24 | 0.84% |

*3.5. Assessment of Linearity of qPCR and ddPCR*

To assess the linearity of the ddPCR quantification, the log copy number data against log dilution were fitted to a linear regression model. Serially diluted pTOPO-WSMoV and pTOPO-MYSV plasmids exhibited good linearity in qPCR and ddPCR assays. The dynamic range of the ddPCR-based assay for MYSV and WSMoV was good in ddPCR (Figure 4A,B). Moreover, the standard curves of WSMoV and MYSV in ddPCR had regression of $Y = 1.0167X - 0.0425$ ($R^2 = 0.9998$) and $Y = 0.9928X + 0.0186$ ($R^2 = 0.9996$), respectively (Figure 4C). In qPCR, the standard curves of WSMoV and MYSV had regression of $Y = 1.0469X - 0.1655$ ($R^2 = 0.9971$) and $Y = 1.0579X - 0.1808$ ($R^2 = 0.9981$), respectively (Figure 4D). The results indicated that the linear regression model exhibited a good linear correlation with the varying concentrations of targets.

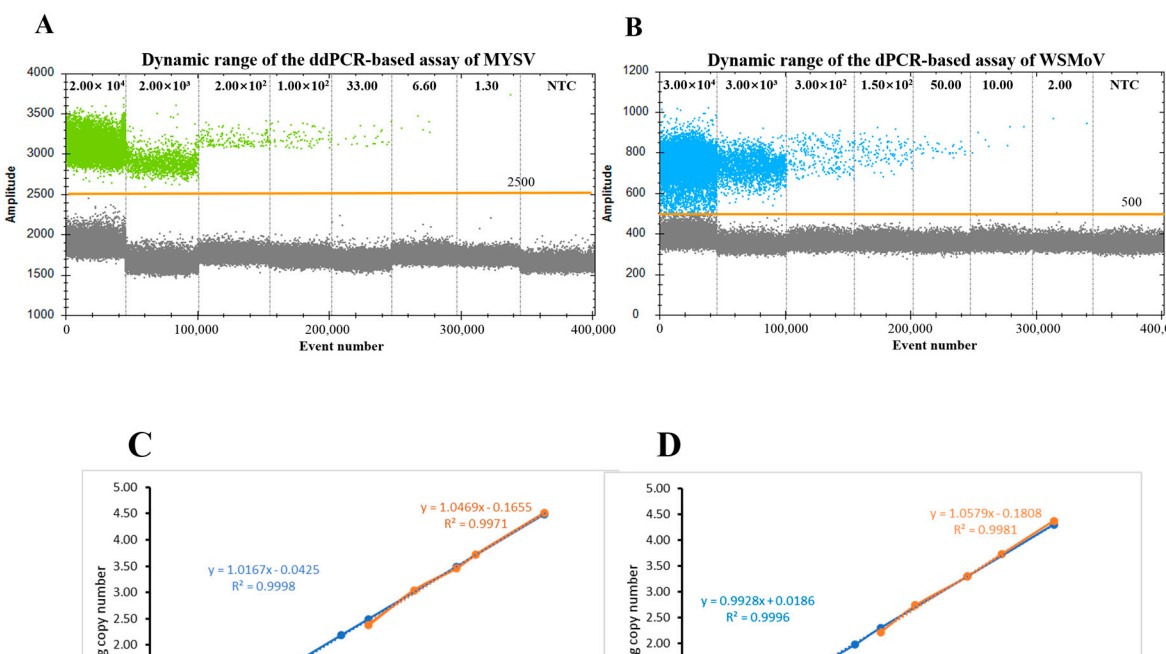

**Figure 4.** Analytical linearity of pTOPO−WSMoV and pTOPO−MYSV plasmids under ddPCR and qPCR. (**A**) Dynamic range of MYSV under ddPCR. (**B**) Dynamic range of WSMoV under ddPCR. (**C**) The pTOPO−WSMoV plasmid was used to construct the standard curves under ddPCR and qPCR. (**D**) The pTOPO−MYSV plasmid was used to construct the standard curves under ddPCR and qPCR. The qPCR and ddPCR results are indicated in orange and blue, respectively.

### 3.6. Assessment of Detection Limits of qPCR and ddPCR

For WSMoV, the detection limit was 18 copies/reaction for ddPCR and 225 copies/reaction for qPCR (Table 2). The detection limit was approximately 12−times higher for ddPCR than for qPCR. Similarly, for MYSV, the detection limit was 9 copies/reaction for ddPCR and 168 copies/reaction for qPCR. It was approximately 18−times higher for ddPCR than for qPCR.

**Table 2.** Detection limits of quantitative real-time PCR (qPCR) and droplet digital PCR (ddPCR).

| Name | Input of Plasmid Copy Number | qPCR Hit Rate (Positive/Total) | dPCR Hit Rate (Positive/Total) |
|---|---|---|---|
| WSMoV | 500 | 1.00 (24/24) | ND |
|  | 300 | 0.96 (23/24) | ND |
|  | 200 | 0.88 (21/24) | ND |
|  | 100 | 0.71 (17/24) | ND |
|  | 50 | ND | 1.00 (24/24) |
|  | 20 | ND | 0.96 (23/24) |
|  | 10 | ND | 0.92 (22/24) |
|  | 5 | ND | 0.75 (18/24) |
|  | 2 | ND | 0.38 (9/24) |
|  | 1 | ND | 0.13 (3/24) |
|  | NTC | 0.00 (0/24) | 0.00 (0/24) |
|  | LOD | 225 | 18 |

**Table 2.** *Cont.*

| Name | Input of Plasmid Copy Number | qPCR Hit Rate (Positive/Total) | dPCR Hit Rate (Positive/Total) |
|---|---|---|---|
| MYSV | 500 | 1.00 (24/24) | ND |
| | 300 | 1.00 (24/24) | ND |
| | 200 | 0.96 (23/24) | ND |
| | 100 | 0.83 (20/24) | ND |
| | 50 | ND | 1.00 (24/24) |
| | 20 | ND | 1.00 (24/24) |
| | 10 | ND | 0.96 (23/24) |
| | 5 | ND | 0.92 (22/24) |
| | 2 | ND | 0.38 (9/24) |
| | 1 | ND | 0.13 (3/24) |
| | NTC | 0.00 (0/24) | 0.00 (0/24) |
| | LOD | 168 | 9 |

NTC: no-template control, ND: No detection.

*3.7. RT-dPCR Detection of Field Samples of WSMoV and MYSV*

To assess the performance of RT-dPCR for detecting WSMoV and MYSV, 62 infected samples of melon, watermelon, and weed were collected from fields. At the same time, they were subjected to the RT-qPCR assay, and the results of both were compared (Table 3). For all 62 samples, the positivity rate for WSMoV was 91.94% (57/62) under RT-dPCR and 59.68% (37/62) under RT-qPCR. The positive rate for MYSV was 51.61% (32/62) under RT-dPCR and 43.39% (30/62) under RT-qPCR. The results indicated that the RT-dPCR assay was more accurate than the RT-qPCR assay.

**Table 3.** Detection of WSMoV and MYSV in field samples by RT-qPCR and RT-ddPCR.

| | WSMoV | | | MYSV | | |
|---|---|---|---|---|---|---|
| | Positive | Negative | Total | Positive | Negative | Total |
| qPCR | 37 | 25 | 62 | 30 | 32 | 62 |
| ddPCR | 57 | 5 | 62 | 32 | 30 | 62 |

**4. Discussion**

WSMoV and MYSV cause serious losses in cucurbit production. Both viruses depend on thrips (*T. palmi*) for their transmission to host plants [28]. These viruses cause yellowing and withering of hosts, as well as loss of commercial value of fruits. Currently, the diagnosis and management of these two viruses mainly focus on the early detection and monitoring of the disease. Therefore, the diagnostic technique must be rapid, reliable, and adaptable to all types of watermelon, melon, and weeds. In this study, we developed and applied ddPCR to detect WSMoV and MYSV.

Here, the primers and TaqMan probes targeting the S RNA segments of the *N* gene in both WSMoV and MYSV (See Supplementary Sequences 1 and Figure S1) exhibited excellent specificity and were suitable for the detection of two viruses using ddPCR. WSMoV and MYSV are from the genus *Tospovirus*; the genomes of WSMoV and MYSV consist of three parts, including small (S), medium (M), and large (L) segments [4]. The S segment is composed of two parts, namely, nucleocapsid (N) protein and nonstructural protein (NSs). The *N* gene is the basis for the virus species of the genus, and it is believed that the sequence identity reaches more than 90% for the same virus [29,30]. The specificity assessments of the ddPCR assay for MYSV and WSMoV revealed that the primers and probes targeting the *N* gene in this study were very species specific; therefore, the primers and probes for the *N* gene of both WSMoV and MYSV were suitable and could be used for the detection of targets of the two viruses.

In this study, RT-dPCR was observed to be an accurate and highly sensitive method for the absolute quantification of WSMoV and MYSV in melon, watermelon, and weed samples. Some weeds were asymptomatic sources. It is necessary to screen the asymptomatic healthy-appearing weeds both in the crop field and at the edges. In addition, less sensitive methods may not detect low virus concentration in some plants. In the field, when thrips are found and the hosts have not yet exhibited symptoms, RT-dPCR may be a reliable and highly sensitive method to detect the virus accumulation in *T. palmi*. It may be necessary and important to rapidly detect the viruses for early warning. Recently, several studies indicated that ddPCR has more specificity and sensitivity of detecting low viroid loads than qPCR [31,32]. In this study, ddPCR was more suitable for the detection of WSMoV and MYSV, with the detection limits higher by approximately 12- and 18-times than those under qPCR, respectively. For the field samples, the positivity rate was 91.94% and 51.61% under ddPCR and 59.68% and 43.39% under qPCR for WSMoV and MYSV, respectively. The results indicated that RT-dPCR had a notably higher accuracy and detection rate than RT-qPCR. The results are consistent with previous studies. The sensitivity of ddPCR was higher by approximately 10-fold than that of qPCR for the detection of apple scar skin viroid, when the viral load was very low [33]. For the samples infected with group A porcine rotavirus (PoRVA), the positivity rate under ddPCR and qPCR was 5.6% and 4.4%, respectively [26]. Therefore, the ddPCR methods described in this study can widely detect viruses from field samples and can be used in the clinical diagnosis of WSMoV and MYSV infections.

Furthermore, RT-dPCR and RT-qPCR exhibited a good degree of linearity and reproducibility in this study. However, the reproducibility of ddPCR was slightly better than that of the qPCR assay. The optimized ddPCR assay exhibited good reproducibility for detecting WSMoV and MYSV. Furthermore, the RT-dPCR and RT-qPCR assays exhibited a high degree of linearity and quantitative correlation. RT-dPCR displayed a relatively more wide-ranging dynamic range in this study. All samples were collected from the fields four times and from four provinces; at that time, the temperature was approximately 35 °C, which might reduce the symptoms and viral load. Moreover, the samples included asymptomatic leaves. Therefore, the results indicated that RT-dPCR is a more sensitive method for detecting WSMoV and MYSV, particularly with extremely low viral loads during latent periods of infection and high temperatures. It can detect 8 copy/μL of MYSV, which is similar to detecting 8 and 9 copies/μL of ZYMV and WMV, respectively [24]. The optimized method in this study was successfully applied on the plants and weeds from fields. In future, this method could be used in thrips to monitor these viruses because they are mainly transmitted by thrips, which are very small in size (approximately 0.5–2 mm). The method is effective for the early monitoring of disease.

In conclusion, RT-dPCR tests targeting WSMoV and MYSV were developed in this study. The primers and probes of both WSMoV and MYSV exhibited excellent specificity and suitability. The ddPCR method exhibited good specificity, linearity, and reproducibility. The method was reliable and accurate across a wide range of plants, including watermelon, melon, and weeds. The early and accurate detection of MYSV and WSMoV will help in assessing the occurrence of epidemics of both viruses and reducing the prevalence of MYSV and WSMoV infections. This study provided an important basis for developing integrated management strategies for both viruses.

**Supplementary Materials:** The following supporting information can be downloaded at: https://www.mdpi.com/article/10.3390/horticulturae10030199/s1. Figure S1: 1.0% electrophoresis gel of agarose; Table S1: the primers used for detecting the virus; Supplementary Sequences 1: Watermelon silver mottle virus (WSMoV) N gene; Melon yellow spot virus (MYSV) N gene.

**Author Contributions:** H.W., experimental design; M.L., writing—original draft; W.L. and J.X., the experiments in lab; M.W. and Y.H., resources; B.K. and L.L., writing—review and editing; B.P. and Q.G., funding acquisition. All authors have read and agreed to the published version of the manuscript.

**Funding:** This work was supported by grants from the Opening Project Fund of Key Laboratory of Vegetable Biology of Hainan Province (HAAS2023PT0209), China Agriculture Research System of MOF and MARA (CARS-25), the Agricultural Science and Technology Innovation Program (CAAS-ASTIP-2022-ZFRI-09), the National Natural Science Foundation of China (U21A20229), and the Science and Technology Major Project of Xinjiang Uygur Autonomous Region, China (2023A02009).

**Data Availability Statement:** Data are contained within the article.

**Conflicts of Interest:** The authors declare no conflicts of interest.

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
