# Peer review of "Development and Application of Droplet Digital PCR Assay for the Detection of Watermelon Silver Mottle Virus and Melon Yellow Spot Virus"

_horticulturae, doi:10.3390/horticulturae10030199_

Round 1
Reviewer 1 Report
Comments and Suggestions for Authors
Watermelon silver mottle virus (WSMoV) and melon yellow spot virus (MYSV) pose significant threat to watermelon, melon, and cucumber in China. Developing and applying ddPCR for epidemiological surveillance may provide another important tool. The manuscript is well-organized and easy to follow. Yet, upon the data provided, it is insufficient to claim some corresponding conclusions. Below, please finds two of my modest suggestion.
1, The authors did a very detailed work in optimizing the ddPCR pipeline. Yet, the authors did not provide convincing reasons that N gene was selected. As claimed in line 291, “N gene of both WSMoV and MYSV exhibited excellent specificity.” For virus detection, generally, several primers should be designed and compare their sensitivity. With that, the limit of ddPCR or qPCR might improve.
2, The authors optimized the pipeline for ddPCR. However, the optimizing data for qPCR is missing. It is possible that the best annealing temperature and probe/primer ration for ddPCR and qPCR is different, due to buffer variation. It would be good if qPCR data would be included. Or it would be tough to claim in the abstract that “The detection limits of WSMoV and MYSV in ddPCR were 18 and 9 copies/μL and were approximately 12 and 18 times more than those in qPCR, respectively”. It is widely known that ddPCR has a higher sensitivity compared to qPCR. Yet, when it comes to the times difference, it should be very cautious.
3, Minor suggestions:
In Table 2, please provide the full name for “ND”.
Table 3 is a bit confusing. Please divided it into two tables.
Comments on the Quality of English Language
Minor editing of English language required. Typos were detected. For example, in line 59, "transmitted by" should not be italic.
Author Response
Responses to reviewers:
Our deepest gratitude goes to you for your careful work and thoughtful suggestions that have helped improve this manuscript substantially. We have carefully addressed all your suggestions and concerns. Please see below our replies. We hope the reviewers are satisfied with our revised manuscript. The responses to the reviewers’ comments are presented below.
Open Review1
Comments and Suggestions for Authors:
Watermelon silver mottle virus (WSMoV) and melon yellow spot virus (MYSV) pose significant threat to watermelon, melon, and cucumber in China. Developing and applying ddPCR for epidemiological surveillance may provide another important tool. The manuscript is well-organized and easy to follow. Yet, upon the data provided, it is insufficient to claim some corresponding conclusions. Below, please finds two of my modest suggestion.
1, The authors did a very detailed work in optimizing the ddPCR pipeline. Yet, the authors did not provide convincing reasons that N gene was selected. As claimed in line 291, “N gene of both WSMoV and MYSV exhibited excellent specificity.” For virus detection, generally, several primers should be designed and compare their sensitivity. With that, the limit of ddPCR or qPCR might improve.
Response from the authors: Thank you for pointing this question and providing advice, we answer the questions in two parts.
- Species demarcation relies on the N protein sequences encoded by N gene with orthotospoviruses, N gene was specificity to orthotospoviruses, so we chose the N gene to detect WSMoV and MYSV, we added the references in manuscript.
A distinct species within the genus, an isolate’s N protein sequence should show less than 90% amino acid identity with that of any other tospovirus species(Oliver etal, 2016). Species demarcation relies on amino acid (aa) sequence identity (<90%) of one of its major structural proteins, i.e., the nucleo(capsid) protein (N), with all other established orthotospoviruses. (Kormelink etal, 2021).
Oliver, J.E.; Whitfield, A.E. The Genus Tospovirus: Emerging Bunyaviruses that Threaten Food Security. Annual Review of Virology 2016, 3, annurev-virology-100114-055036.
Kormelink, R.; Verchot, J.; Tao, X.; Desbiez, C. The Bunyavirales: The Plant-Infecting Counterparts. Viruses 2021, 13, 842, doi:doi: 10.3390/v13050842.
- About the primers and probes, we designed two pairs of primers and probes for each virus at that time, and conducted pre-experiments, and found that there was a suitable pair, so we chose it without using the other one, we did not show them in the manuscript. The primers we designed are in the table below.
Name |
Sequences |
|
WSMoV-ddF1 |
TGAATGTCCAGTCCTCAGGT |
|
WSMoV-ddR1 |
GCCTGCAAGAGCGGTAAGTA |
|
WSMoV-rpob1 |
CCGACGTCAACACTTGTGGCAACA |
|
WSMoV-ddF2 |
ACTTCTCCACCTACGAGCAA |
good |
WSMoV-ddR2 |
AGCCTGTTTGAACATAACATCCA |
|
WSMoV-ddprob2 |
TGTGAAGCAGACAGAACCTTAGCCACT |
|
MYSV-ddprob1 |
ATAGGCCAAAGGCAGGACAA |
|
MYSV-ddF1 |
GGAGGACCTCTGTCTTTGCT |
|
MYSV-ddR1 |
TGCTAGTCTGCATTCGCTCTGCCC |
|
MYSV-ddF2 |
TCACCTCATCTCTCACATCCG |
good |
MYSV-ddR2 |
ACACCATGTCTACCGTTGCT |
|
MYSV-ddprob2 |
TCCGACTTGCCACCACTGAGAAGT |
2, The authors optimized the pipeline for ddPCR. However, the optimizing data for qPCR is missing. It is possible that the best annealing temperature and probe/primer ration for ddPCR and qPCR is different, due to buffer variation. It would be good if qPCR data would be included. Or it would be tough to claim in the abstract that “The detection limits of WSMoV and MYSV in ddPCR were 18 and 9 copies/μL and were approximately 12 and 18 times more than those in qPCR, respectively”. It is widely known that ddPCR has a higher sensitivity compared to qPCR. Yet, when it comes to the times difference, it should be very cautious.
Response from the authors: Thank you for raising this question.
In the experiment, we also optimized qPCR, we found that the annealing temperature and probe/primer ration for ddPCR were also suitable for qPCR. In addtion, here we mainly detected the viruses by ddPCR, in our oppion, the optimized data for qPCR is not shown.
3, Minor suggestions:
In Table 2, please provide the full name for “ND”.
Response from the authors: Thank you for advice, ND is abbreviated for no detection, we added it in manuscript.
Table 3 is a bit confusing. Please divided it into two tables.
Response from the authors: Thanks for your suggestion. We revised the table 3 as below, the table is clear.
Table 3.
WSMoV |
MYSV |
|||||
Positive |
Negative |
Total |
Positive |
Negative |
Total |
|
qPCR |
37 |
25 |
62 |
30 |
32 |
62 |
ddPCR |
57 |
5 |
62 |
32 |
30 |
62 |
Comments on the Quality of English Language
Minor editing of English language required. Typos were detected. For example, in line 59, "transmitted by" should not be italic.
Response from the authors: Thank you for your question. We performed language revision of the full manuscript.

Reviewer 2 Report
Comments and Suggestions for Authors
Overall, fine written article. While reading feels a little bit overdesigned and complicated. Simple ddPCR method could have been made and optimised by much smaller group of people. It is not that much necessary to compare it to qPCR. It is obvious that ddPCR is more accurate and detects smaller amounts of target nucleic acid. It is not a surprise. You try present your ddPCR as miracle method that is better than everything else. Nothing in the world is like that. There is always positives and negatives. And you avoid talking about drawbacks of your method. There are different methods and they are used in “the right tool for the given job” approach. ddPCR will not make conventional simple PCR or qPCR obsolete. As illumina sequencing did not make the sanger sequencing obsolete.
So the use of method could be presented more accurately. What exactly are its plusses and minuses. And the disease control strategies are very vague terms and need more precise explanation. How are you preventing the infections after you detected them.
List of things to fix:
Line 2: “double digital” - change to droplet digital
Line 16: could you elaborate what “disease management” means? You detect virus early on, then what? Do you at least actually destroy the sources of infection, or just “monitor” it.
Line 17: “reqired” - required
Line 18: “N gene” – this is first mention, thus you should write its full name (nucleocapsid gene)
Line 29: “be widely used” – I disagree with liberal use of such claim. ddPCR is EXPENSIVE, requiring two devices that most likely costs way above 50000 dolars. (delete “widely”)
Line 34: elaborate “developing integrated management strategies for these viruses”. Or don’t use vague terms. Maybe something like “fast extermination of detected infection sources” sounds more direct and truthful?
Line 36: no gap in “PCR;detection”
Line 50: “Taiwa” – Taiwan?
Line 59: “transmited by” is in italics font
Line 63: “development of virus detection technology should be promoted” – this sounds as there are no detection methods and you are developing one. Those are not new unknown viruses, there already are methods to detect and identify them. More accurate would be to say that you are improving their detection.
Line 77: “they need more time and cost” – this is not exactly true. Looking at one sample cost, oxford nanopore will cost you 1k (MiniON is cheap) and give you sequencing of all rnas in sample. AND will do that overnight! So it is not slow. While for ddPCR you first need to buy droplet gen and analyser which will be >50k to do that one sample. In truth, those are incomparable methods. You are comparing a nucleotide sequencing device with a PCR machine…
Line 78: “more accurate” – again, in general, without context, is not true. PCR is NOT more accurate than high throughput sequencing. Problem is, you mentioned a lot of very different methods (different pcrs, elisa, sequencing) and are trying to bunch them up together and say that your ddpcr is magical method that is better than all of them. Which is not, because your method is expensive and won’t be accessible to everyone and simpler more primitive methods will always remain as cheap alternatives.
Over whole article ddPCR is oversold as panacea. While you should describe its niche use more precisely. And maybe this paragraph here, where you mentioned all other available methods is a good place to do so. It is very expensive. It requires two specific expensive devices. It will require additional costs of taqman probes in each reaction, compared to simple RT-PCR. So don’t lie to the reader that its higher sensitivity has no drawbacks. Describe better where it can be used and why that cost is justified. Why higher sensitivity is even needed? There are many plant pathology problems missing from your article’s introduction and/or discussion sections that could been explained to the reader. Like the weeds as asymptomatic sources of infection. The need of screening of asymptotic healthy looking weeds both in the crop field and at the edges. The low virus concentration in some plants that can avoid detection by less sensitive methods. That insects keep feeding and spreading the virus from asymptotic plants again and again multiplying the problem.
Line 109: I don’t se the point of citation nr 33. Why is it needed? You did your research, we see the results, we don’t need proof that you collected some plant samples. Also it is next to ToLCNDV but there is no citation next to CMV, so why give proof to one virus samples but not to the others.
Line 112, 113, 115: (tiangen, china) – maybe this journal has different rule, but in general, usually when first mentioning manufacturer you give its name and country and city of origin. When you mention it second time you only give its name and skip its origin place. In next paragraphs, lines 125, 138 you mention manufacturers differently. Chose a system and stick to it, make everywhere in same format.
Line 121: unnecessary gap “AGGG TTAC”
Line 134: gap “CACC TCAT”
Line 289: “menlon” – melon?
Line 333: “viruse” – virus?
Line 335: elaborate “reduce the occurrence of disease”. How exactly it will be reduced? What steps there are in this chain of actions?
Line 345: Author contributions look a bit vague. It is not clear why so many people where needed in this study. I would partition and mention more separate tasks, like maybe some people did the droplet pcr, another people did qpcrs, who collected the samples and so on.
Author Response
Cover letter to editor:
Dear editor,
Our deepest gratitude goes to you for your careful work and thoughtful suggestions that have helped improve this manuscript substantially. We have carefully addressed all your suggestions and concerns. Please see below our replies one by one. We hope that you are satisfied with our revised manuscript. The responses to comments are presented below in red color. Thanks a lot!
Best wishes!
Huijie wu
- The article title is different from the one submitted online at susy.mdpi.com. Please confirm which one is correct.
Thanks a lot, reviewer 2 advised us to chang “double digital” to “droplet digital”, so we revised it.
- Please carefully check the accuracy of names and affiliations.
We had carefully checked the accuracy of names and affiliations.
- The affiliation numbers should appear in numerical order. Please check the suggested changes.
We revised them in red color.
- The name of this author is different from the one submitted online at susy.mdpi.com. Please confirm which one is correct.
We confirmed all names.
- We deleted the province name from affiliations 1 and 3. Please confirm
That is OK
- We added the email addresses here according to those submitted online at susy.mdpi.com. Please confirm.
That is OK
- Please make sure that the address information is arranged from subordinate to superior.
We revised them in red color.
- Please check all author names carefully.
We had carefully checked all author names.
- Please state the version number of the software.
Here Excel 2016 Software was used, we added it in manuscript.
- We moved all figures/tables behind their first citations and deleted section 3.8. Please confirm
It is OK.
- Please use commas to separate thousands for numbers with five or more digits (not four digits) in the picture, e.g., “10000” should be “10,000”. Figure 1
We revised the number “10000” into “10,000” in Figure 1.
- Please use commas to separate thousands for numbers with five or more digits (not four digits) in the picture, e.g., “10000” should be “10,000”. Figure 2
We revised the number “10000” into “10,000” in Figure 2.
- Figure 3
Please add explanations for subfigures A and B, please use commas to separate thousands for numbers with five or more digits (not four digits) in the picture, e.g., “10000” should be “10,000”.
We added explanations for subfigures A and B, and changed the “10000” into“10,000” in Figure 3.
- Figure 4.
Please change the terms into scientific notations in the figure, e.g., “8 × 103”, not “8E3”. Please use commas to separate thousands for numbers with five or more digits (not four digits) in the picture, e.g., “10000” should be “10,000”.
Please change the hyphen (-) into a minus sign (−, “U+2212”), e.g., “-1” should be “−1”.
We revised the figure “E3” into “× 103” , changed the “10000” into“10,000” in Figure 4,
and changed the hyphen into a minus sign.
- Table 2. Could it be deleted? There is no * in the table body
- * could be deleted.
- Table 3. Please confirm if the italics is unnecessary and can be removed. The following highlights are the same
Thanks a lot. We revised all the contents in manuscript.
- Information regarding the funder and the funding number should be provided. Please check the accuracy of funding data and any other information carefully.
We carefully checked the accuracy of funding data, and changed the location between the Opening Project Fund of Key Laboratory of Vegetable Biology of Hainan Province (HAAS2023PT0209), China Agriculture Research System of MOF and MARA (CARS-25).
Data Availability Statement: Not applicable. No new data were created or analyzed in this study.
- We added stage of publication, please confirm.
Thanks a lot. We revised all the references in manuscript.
- Article/volume/page number is added, please confirm. The following highlights are the same.
Thanks a lot. We revised all the references in manuscript.
Dear Academic Editor,
Our deepest gratitude goes to you for your careful work and thoughtful suggestions that have helped improve this manuscript substantially. We have carefully addressed all your suggestions and concerns. Please see below our replies one by one. We hope that you are satisfied with our revised manuscript. The responses to comments are presented below in red color. Thanks a lot!
Best wishes!
Huijie wu
Dear Authors, the work describes the set up of two protocols to detect watermelon silver mottle virus and melon yellow spot virus in cucurbits. The set up procedure for probe and primer design is accurate and the new methods could be useful for the early detection of the two viruses in the field or in greenhouse. On the other side, reference strains of the target (and non target) viruses or reference analytical methods are not mentioned and cross reaction with other tosporiviruses were not verified (mainly, in the weeds) Informations on these points should be provided to allow the acceptance of the manuscript.
Response from the authors: Thank you for raising this question, it is very good question. We answerd this question in three parts.
- About the reference strains of the target :
Based on the sequences submitted in the GenBank for WSMoV(MT226398.1, NC_003843.1 X78556.1) and for MYSV (KX711613.1, AY673635.1, AY673636.1), we designed primers to amply the N genes. The samples collected from different areas were used to be analyzed, one sample was selected for each region. RT-PCR was used to amplify the N gene of WSMoV and MYSV, respectively. The probes and primers were designed based on the obtained sequences. We provide the sequences of WSMoV and MYSV as below.
> Watermelon silver mottle virus (WSMoV)
GCTGTTCCAGGGTTACTTTCACTGAGTATTTTGACAGCCTGTTTGAACATAACATCCAAATCATCCTTAAATTCAACCTGTGAAGCAGAAAGAACCTTAGCCACCTTACAAACTTGCTCATAGGTGGAGAAGTTCTTTATACCCAACTTTTCTTTCTTTATATTTTGGTAATAAGCTAAAGGGAAAATAATCGGTGCCAGTCCCCTTATACTAGATAAAAGAGGTAGAGGTCCTCCAATACATAACATCATCCTCAAGGCACAAGAATCATAAGATGCGGGCACATTCAACCCATAAGCAGCAACTAATGGTAGTTCCATGATTTTCCCATACATTTCCTGTCTAGCAGCTTCATTCTTGCTTTTCTCTACCATACTAACTATTTTTGTCCTGATAAAAGCTTCTGTCCTCTTGAATGTCCAGTCCTCAGGTCCGACGTCAACACTTGTGGCAACAATATTTTTACCGCAAAAATTATACTTACCGCTCTTGCAGGCTGCAAAAATCTGTTTCCTGCATTTTAGGATATTCAAGCAGTTTGTAAAGGTTATCTCTATATTTTTGTTGTTGTAAAAAAAAGCTTTGAAGCTAAATCCGGGAGTGGAGTCC
> Melon yellow spot virus (MYSV)
TTAAACTTCAATGGACTTAGATCTGGAAGAAGAAGCTTTACTCTTAGAAGAGCTCTCACCCATTCCATCAACTACAAGAGAGGATTTGAAAGCCTGTTCCATGAATTGCACCTGTTCATTGTAAATCTTCAGGGATATGCCACTGGTTGTTCCTGGTTTGCAGTCAGCCAAGATCTTCACACAACTTTTGAAAAGTTCATCAAATTCTTTCTTGAAGGTCATATTGCTAGCAGACATAACGCGAGCAATTTTGCAGATCTGCTCATATGTAGAGAAATTCTTTATTCCTAATTGCTCTTTCTTGACATTCTGAAAATAGGCCAAAGGCAGGACAACAGGGCAGAGCGAATGCAGACTAGCAAGCAAAGACAGAGGTCCTCCAATGCATAGCATTAATCTCAAAGCTGTCATGTCAAACTTTGCAGGCACAGTTAAACCATATGCAGCAACCAATGGAAGCTGCATGGCTTTCTCATACATTTTCTGCTTTTCTTCCTCATTCTCTGTTTTTTCTGCAATGCTGATCATCAAAGTTCTGATAACAGCTTCAGTTCGTTTAAATGTCCAATCATTAGGACCAACATTATCCCCAGATGCAACTATCTTTTTCCCGCAAAACTTGTAGTTTCCAGATTTGCAAGCTGCATAAACTTGTTTTCTGCTTTTCAGAATAGTTATACCGTTATTATAATTTAATTTCACCTCATCTCTCACATCCGTAAAGAATGAATGGAAGTTAAATCCTTCGGCTGATTCCTCTGTTTCTATTTCAACTTCCGACTTGCCACCACTAAGAAGTTCTTGGATTTTCTCCTTTGTCAGCTTAGCAACGGTAGACATGGTGTTTACTTAGGAAACGTGTACTTTGACTTGCTGATGTTGAATT
- About reference analytical methods
I don't understand the meaning of this sentence very well. Do you want us to give the results of other methods? If that is the case, let us talk about problem. In our oppion, other methods may also be good, but untill now there is no method for digital PCR to detect two viruses. Not only that, when the virus accumulation is very low in plants, the methods such as reverse transcriptase polymerase chain reaction (RT-PCR), enzyme-linked immunosorbent assay (ELISA), western blotting, and double-antibody sandwich ELISA (DAS-ELISA) may be not used to detect the virus because of the method detection limits. Though high throughput sequencing (HTS) may detect a lower virus accumulation, it needs more time and cost than dPCR, and HTS requires specialized machines which are expensive. Therefore, here we established ddPCR a further method to detect two viruses, other methods are not shown here.
- Other tosporiviruses were not verified Informations.
Thank you for raising this question and giving the advice, this is a good question. Unfortunately, we had no the informations about other tosporiviruses.
We searched some data, although this information is not updated very timely, but it also gives us some advise. Based on the Descriptions of Plant Viruses (https://www.dpvweb.net/dpv/taxon-index), we known that some viruses belonging to tosporiviruses genus do not occur in China, so we maybe need more time to get reference strains of the target. We will continue to do this work in future.
Virus |
Geographical distribution |
Host range |
Vector |
Definitive species |
|||
Tomato spotted wilt virus (TSWV)17 |
worldwide10 |
Very broad (800 species) |
Frankliniella fusca 23 |
Impatiens necrotic spot virus (INSV)5, 15 |
USA15, Europe5, 16, 27 |
Mainly ornamentals, e.g. |
F. occidentalis 3, 29 |
Tomato chlorotic spot virus (TCSV)6 |
Brazil4 |
Tomato |
F. intonsa 30 |
Groundnut ringspot virus (GRSV)6 |
Argentine7, Brazil4, South Africa |
Groundnut, tomato |
F. occidentalis 30 |
Watermelon silver mottle virus (WSMV)13, 31 |
Japan14, Taiwan13, 31 |
Watermelon, other cucurbits |
T. palmi 32 |
Groundnut* bud necrosis virus (GBNV)21, 25 |
India21, South-East Asia |
Groundnut |
T. palmi 18, 28 |
Iris yellow spot virus(IYSV)2 |
The Netherlands |
Iris |
unknown |
Melon spotted wilt virus(MSWV)34 |
Japan34 |
Melon |
T. palmi |
Groundnut* yellow spot virus (GYSV)20 |
India, Thailand20 |
Groundnut |
unknown |
Isolate Chry1 (no species name coined yet)22 |
Brazil22 |
Chrysanthemum, tomato |
unknown |
Isolate BR-09Z22 (no species name coined yet) |
Brazil22 |
Zucchini |
unknown |
See comment below:
Lines 20-21: Although both qPCR and ddPCR exhibited good reproducibility, here ddPCR was better than qPCR, the reproducibility of ddPCR was proved to be 100%. ("reproducibility" is written 3 times in two lines. Please revise the sentence)
Response from the authors: Thanks a lot! We changed the sentences: Although both qPCR and ddPCR exhibited good reproducibility, here ddPCR was better than qPCR, the reproducibility of ddPCR was proved to be 100%.
Line 30: replace “clinical” with “analytical”
Response from the authors: Thanks, we changed “clinical” to “analytical” in manuscript.
Line 46: what the authors mean with “our…. Survey”. Which survey?
Response from the authors: Thanks, we have investigated the incidence of viral diseases in the watermelon and melon field almost every year since 2020.
Line 58: “Chieh-Qua”. Add scientific name of the host
Response from the authors: Thanks, we added scientific name Benincasa hispida Cogn. var. chieh-qua in manuscript.
Line 70: which limitation? Which are the limitation addressed and solved in this work?
Response from the authors: Thanks a lot. When virus accumulation is very low in some plants, the virus could avoid detection by less sensitive methods. We added the sentences in manuscript.
Lane 91: I suggest to replace “another” with “a further”
Response from the authors: Thanks, we changed “another” to “a further” in manuscript.
Lines 115-118: N gene of WSMoV and MYSV was amplified (primers WSMoV-S-F: 5ʹ-GCTGTTCCAGGGTTACTTTC-3ʹ/WSMoV-S-R: 5ʹ-GGACTCCACTCCCGGATTTA-3ʹ and MYSV-cp-F: 5ʹ-TTAAACTTCAATGGACTTAGATC-3ʹ/MYSV-cp-R: 5ʹ-ATTCAACATCAGCAAGTCAA-3ʹ), which yielded 609- and 886-bp amplicons, respectively. Could be revised as follows: “N gene of WSMoV and MYSV was amplified with primers pairs WSMoV-S-F: 5ʹ-GCTGTTCCAGGGTTACTTTC-3ʹ/WSMoV-S-R: 5ʹ-GGACTCCACTCCCGGATTTA-3ʹ and MYSV-cp-F: 5ʹ-TTAAACTTCAATGGACTTAGATC-3ʹ/MYSV-cp-R: 5ʹ-ATTCAACATCAGCAAGTCAA-3ʹ, which yielded 609- and 886-bp amplicons, respectively.
Response from the authors: Thanks, we revised it in manuscript.
Lines 155-157: specifity was evaluated on viral reference strains? How the author detected CGMMV, CCYV, ToLCNDV, and CMV? Reference strains were used for WSMoV or MYSV? Which strains? Did the authors test crossreaction with other tospoviruses?
Response from the authors: Thank you for this question, it is very good. The specifity was evaluated viral reference strains on watermelon and melon. We detect the CGMMV, CCYV, ToLCNDV, and CMV by RT-PCR and sequencing, and stored these viruses in our Lab before.
The sequences of reference strains for WSMoV (MT226398.1, NC_003843.1 and X78556.1) and for MYSV (KX711613.1, AY673635.1 and AY673636.1) were analyzed and the primers were designed based on the obtained N genes sequences. The samples collected from different areas were used, one sample was selected for each region. RT-PCR was used to amplify and sequence the N genes of WSMoV and MYSV, respectively. We provide the sequences of WSMoV and MYSV as above.
Unfortunately, we had no the informations about other tosporiviruses, we will continue to do this work in future.
Lines 287: results on analyses conducted on weeds are mentioned in the discussion but they were not reported in materials and methods or results section. Which weeds? How many samples of weeds were analyzed? How can the author demonstrate that the weeds were really infected?
Response from the authors: Thanks a lot for the questions. Line 185, we added “including watermelon, melon and five weeds”. Two nightshade and three heliotropium indicum samples were collected and detected, WSMoV was not detected, MYSV was detected by two nightshades by RT-dPCR. The RT-PCR was used to amply and sequence N gene for MYSV, the sequences are given as mentioned above.
Figure 1 Nightshade
Figure 2 Heliotropium indicum
Lines 307-314: did the authors compare the r esults otained with qPCR and ddPCR protocols set up in this work with the results from an already published method, preferably validated method? If not, add some sentences in the discussion to explain why they did not and how they can assess that the positive samples are not false positive
Response from the authors: Thank you for raising this question and giving the advice. We do not compare the results otained with qPCR and ddPCR protocols set up in this work with the results from an already published method. We added some sentences line 325-330 to explain the reasons.
Line 326: delete “in”
Response from the authors: Thanks, Line 326 “in” was deleted.
In conclusion, further details (or, preferably, data) on the validation of the protocols have to be provided or, at least, a strong argumentation should be added in the discussion to explain why a detection method of comparison was not used. Finally, if available, provide methods and results of analyses carried out on artificially infected (and non-infected) samples and the codes of reference strain used.
Response from the authors: Thanks a lot. We added some sentences to explain the reasons. When the virus accumulation is very low, the methods such as RT-PCR, ELISA, western blotting, and DAS-ELISA may be not used to detect the viruses because of their detection limits. Though HTS may also detect a lower the virus accumulation, it needs more time and cost than ddPCR, because HTS requires specialized expensive machines. Moreover, RT-dPCR requires virus-specific Taq-man probes, reducing the chance of false positives.
Dear Academic Editor,
Our deepest gratitude goes to you for your careful work and thoughtful suggestions that have helped improve this manuscript substantially. We have carefully addressed all your suggestions and concerns. Please see below our replies one by one. We hope that you are satisfied with our revised manuscript. The responses to comments are presented below in red color. Thanks a lot!
Best wishes!
Huijie wu
Dear Authors, the work describes the set up of two protocols to detect watermelon silver mottle virus and melon yellow spot virus in cucurbits. The set up procedure for probe and primer design is accurate and the new methods could be useful for the early detection of the two viruses in the field or in greenhouse. On the other side, reference strains of the target (and non target) viruses or reference analytical methods are not mentioned and cross reaction with other tosporiviruses were not verified (mainly, in the weeds) Informations on these points should be provided to allow the acceptance of the manuscript.
Response from the authors: Thank you for raising this question, it is very good question. We answerd this question in three parts.
- About the reference strains of the target :
Based on the sequences submitted in the GenBank for WSMoV(MT226398.1, NC_003843.1 X78556.1) and for MYSV (KX711613.1, AY673635.1, AY673636.1), we designed primers to amply the N genes. The samples collected from different areas were used to be analyzed, one sample was selected for each region. RT-PCR was used to amplify the N gene of WSMoV and MYSV, respectively. The probes and primers were designed based on the obtained sequences. We provide the sequences of WSMoV and MYSV as below.
> Watermelon silver mottle virus (WSMoV)
GCTGTTCCAGGGTTACTTTCACTGAGTATTTTGACAGCCTGTTTGAACATAACATCCAAATCATCCTTAAATTCAACCTGTGAAGCAGAAAGAACCTTAGCCACCTTACAAACTTGCTCATAGGTGGAGAAGTTCTTTATACCCAACTTTTCTTTCTTTATATTTTGGTAATAAGCTAAAGGGAAAATAATCGGTGCCAGTCCCCTTATACTAGATAAAAGAGGTAGAGGTCCTCCAATACATAACATCATCCTCAAGGCACAAGAATCATAAGATGCGGGCACATTCAACCCATAAGCAGCAACTAATGGTAGTTCCATGATTTTCCCATACATTTCCTGTCTAGCAGCTTCATTCTTGCTTTTCTCTACCATACTAACTATTTTTGTCCTGATAAAAGCTTCTGTCCTCTTGAATGTCCAGTCCTCAGGTCCGACGTCAACACTTGTGGCAACAATATTTTTACCGCAAAAATTATACTTACCGCTCTTGCAGGCTGCAAAAATCTGTTTCCTGCATTTTAGGATATTCAAGCAGTTTGTAAAGGTTATCTCTATATTTTTGTTGTTGTAAAAAAAAGCTTTGAAGCTAAATCCGGGAGTGGAGTCC
> Melon yellow spot virus (MYSV)
TTAAACTTCAATGGACTTAGATCTGGAAGAAGAAGCTTTACTCTTAGAAGAGCTCTCACCCATTCCATCAACTACAAGAGAGGATTTGAAAGCCTGTTCCATGAATTGCACCTGTTCATTGTAAATCTTCAGGGATATGCCACTGGTTGTTCCTGGTTTGCAGTCAGCCAAGATCTTCACACAACTTTTGAAAAGTTCATCAAATTCTTTCTTGAAGGTCATATTGCTAGCAGACATAACGCGAGCAATTTTGCAGATCTGCTCATATGTAGAGAAATTCTTTATTCCTAATTGCTCTTTCTTGACATTCTGAAAATAGGCCAAAGGCAGGACAACAGGGCAGAGCGAATGCAGACTAGCAAGCAAAGACAGAGGTCCTCCAATGCATAGCATTAATCTCAAAGCTGTCATGTCAAACTTTGCAGGCACAGTTAAACCATATGCAGCAACCAATGGAAGCTGCATGGCTTTCTCATACATTTTCTGCTTTTCTTCCTCATTCTCTGTTTTTTCTGCAATGCTGATCATCAAAGTTCTGATAACAGCTTCAGTTCGTTTAAATGTCCAATCATTAGGACCAACATTATCCCCAGATGCAACTATCTTTTTCCCGCAAAACTTGTAGTTTCCAGATTTGCAAGCTGCATAAACTTGTTTTCTGCTTTTCAGAATAGTTATACCGTTATTATAATTTAATTTCACCTCATCTCTCACATCCGTAAAGAATGAATGGAAGTTAAATCCTTCGGCTGATTCCTCTGTTTCTATTTCAACTTCCGACTTGCCACCACTAAGAAGTTCTTGGATTTTCTCCTTTGTCAGCTTAGCAACGGTAGACATGGTGTTTACTTAGGAAACGTGTACTTTGACTTGCTGATGTTGAATT
- About reference analytical methods
I don't understand the meaning of this sentence very well. Do you want us to give the results of other methods? If that is the case, let us talk about problem. In our oppion, other methods may also be good, but untill now there is no method for digital PCR to detect two viruses. Not only that, when the virus accumulation is very low in plants, the methods such as reverse transcriptase polymerase chain reaction (RT-PCR), enzyme-linked immunosorbent assay (ELISA), western blotting, and double-antibody sandwich ELISA (DAS-ELISA) may be not used to detect the virus because of the method detection limits. Though high throughput sequencing (HTS) may detect a lower virus accumulation, it needs more time and cost than dPCR, and HTS requires specialized machines which are expensive. Therefore, here we established ddPCR a further method to detect two viruses, other methods are not shown here.
- Other tosporiviruses were not verified Informations.
Thank you for raising this question and giving the advice, this is a good question. Unfortunately, we had no the informations about other tosporiviruses.
We searched some data, although this information is not updated very timely, but it also gives us some advise. Based on the Descriptions of Plant Viruses (https://www.dpvweb.net/dpv/taxon-index), we known that some viruses belonging to tosporiviruses genus do not occur in China, so we maybe need more time to get reference strains of the target. We will continue to do this work in future.
Virus |
Geographical distribution |
Host range |
Vector |
Definitive species |
|||
Tomato spotted wilt virus (TSWV)17 |
worldwide10 |
Very broad (800 species) |
Frankliniella fusca 23 |
Impatiens necrotic spot virus (INSV)5, 15 |
USA15, Europe5, 16, 27 |
Mainly ornamentals, e.g. |
F. occidentalis 3, 29 |
Tomato chlorotic spot virus (TCSV)6 |
Brazil4 |
Tomato |
F. intonsa 30 |
Groundnut ringspot virus (GRSV)6 |
Argentine7, Brazil4, South Africa |
Groundnut, tomato |
F. occidentalis 30 |
Watermelon silver mottle virus (WSMV)13, 31 |
Japan14, Taiwan13, 31 |
Watermelon, other cucurbits |
T. palmi 32 |
Groundnut* bud necrosis virus (GBNV)21, 25 |
India21, South-East Asia |
Groundnut |
T. palmi 18, 28 |
Iris yellow spot virus(IYSV)2 |
The Netherlands |
Iris |
unknown |
Melon spotted wilt virus(MSWV)34 |
Japan34 |
Melon |
T. palmi |
Groundnut* yellow spot virus (GYSV)20 |
India, Thailand20 |
Groundnut |
unknown |
Isolate Chry1 (no species name coined yet)22 |
Brazil22 |
Chrysanthemum, tomato |
unknown |
Isolate BR-09Z22 (no species name coined yet) |
Brazil22 |
Zucchini |
unknown |
See comment below:
Lines 20-21: Although both qPCR and ddPCR exhibited good reproducibility, here ddPCR was better than qPCR, the reproducibility of ddPCR was proved to be 100%. ("reproducibility" is written 3 times in two lines. Please revise the sentence)
Response from the authors: Thanks a lot! We changed the sentences: Although both qPCR and ddPCR exhibited good reproducibility, here ddPCR was better than qPCR, the reproducibility of ddPCR was proved to be 100%.
Line 30: replace “clinical” with “analytical”
Response from the authors: Thanks, we changed “clinical” to “analytical” in manuscript.
Line 46: what the authors mean with “our…. Survey”. Which survey?
Response from the authors: Thanks, we have investigated the incidence of viral diseases in the watermelon and melon field almost every year since 2020.
Line 58: “Chieh-Qua”. Add scientific name of the host
Response from the authors: Thanks, we added scientific name Benincasa hispida Cogn. var. chieh-qua in manuscript.
Line 70: which limitation? Which are the limitation addressed and solved in this work?
Response from the authors: Thanks a lot. When virus accumulation is very low in some plants, the virus could avoid detection by less sensitive methods. We added the sentences in manuscript.
Lane 91: I suggest to replace “another” with “a further”
Response from the authors: Thanks, we changed “another” to “a further” in manuscript.
Lines 115-118: N gene of WSMoV and MYSV was amplified (primers WSMoV-S-F: 5ʹ-GCTGTTCCAGGGTTACTTTC-3ʹ/WSMoV-S-R: 5ʹ-GGACTCCACTCCCGGATTTA-3ʹ and MYSV-cp-F: 5ʹ-TTAAACTTCAATGGACTTAGATC-3ʹ/MYSV-cp-R: 5ʹ-ATTCAACATCAGCAAGTCAA-3ʹ), which yielded 609- and 886-bp amplicons, respectively. Could be revised as follows: “N gene of WSMoV and MYSV was amplified with primers pairs WSMoV-S-F: 5ʹ-GCTGTTCCAGGGTTACTTTC-3ʹ/WSMoV-S-R: 5ʹ-GGACTCCACTCCCGGATTTA-3ʹ and MYSV-cp-F: 5ʹ-TTAAACTTCAATGGACTTAGATC-3ʹ/MYSV-cp-R: 5ʹ-ATTCAACATCAGCAAGTCAA-3ʹ, which yielded 609- and 886-bp amplicons, respectively.
Response from the authors: Thanks, we revised it in manuscript.
Lines 155-157: specifity was evaluated on viral reference strains? How the author detected CGMMV, CCYV, ToLCNDV, and CMV? Reference strains were used for WSMoV or MYSV? Which strains? Did the authors test crossreaction with other tospoviruses?
Response from the authors: Thank you for this question, it is very good. The specifity was evaluated viral reference strains on watermelon and melon. We detect the CGMMV, CCYV, ToLCNDV, and CMV by RT-PCR and sequencing, and stored these viruses in our Lab before.
The sequences of reference strains for WSMoV (MT226398.1, NC_003843.1 and X78556.1) and for MYSV (KX711613.1, AY673635.1 and AY673636.1) were analyzed and the primers were designed based on the obtained N genes sequences. The samples collected from different areas were used, one sample was selected for each region. RT-PCR was used to amplify and sequence the N genes of WSMoV and MYSV, respectively. We provide the sequences of WSMoV and MYSV as above.
Unfortunately, we had no the informations about other tosporiviruses, we will continue to do this work in future.
Lines 287: results on analyses conducted on weeds are mentioned in the discussion but they were not reported in materials and methods or results section. Which weeds? How many samples of weeds were analyzed? How can the author demonstrate that the weeds were really infected?
Response from the authors: Thanks a lot for the questions. Line 185, we added “including watermelon, melon and five weeds”. Two nightshade and three heliotropium indicum samples were collected and detected, WSMoV was not detected, MYSV was detected by two nightshades by RT-dPCR. The RT-PCR was used to amply and sequence N gene for MYSV, the sequences are given as mentioned above.
Figure 1 Nightshade
Figure 2 Heliotropium indicum
Lines 307-314: did the authors compare the r esults otained with qPCR and ddPCR protocols set up in this work with the results from an already published method, preferably validated method? If not, add some sentences in the discussion to explain why they did not and how they can assess that the positive samples are not false positive
Response from the authors: Thank you for raising this question and giving the advice. We do not compare the results otained with qPCR and ddPCR protocols set up in this work with the results from an already published method. We added some sentences line 325-330 to explain the reasons.
Line 326: delete “in”
Response from the authors: Thanks, Line 326 “in” was deleted.
In conclusion, further details (or, preferably, data) on the validation of the protocols have to be provided or, at least, a strong argumentation should be added in the discussion to explain why a detection method of comparison was not used. Finally, if available, provide methods and results of analyses carried out on artificially infected (and non-infected) samples and the codes of reference strain used.
Response from the authors: Thanks a lot. We added some sentences to explain the reasons. When the virus accumulation is very low, the methods such as RT-PCR, ELISA, western blotting, and DAS-ELISA may be not used to detect the viruses because of their detection limits. Though HTS may also detect a lower the virus accumulation, it needs more time and cost than ddPCR, because HTS requires specialized expensive machines. Moreover, RT-dPCR requires virus-specific Taq-man probes, reducing the chance of false positives.
Two viruses are thrips-borne and not infected manually hosts, so we cannot provide methods and results of analyses carried out on artificially infected (and non-infected) samples. In our oppion, untill now the infectious clones of two viruses have not yet been constructed in world.
Two viruses are thrips-borne and not infected manually hosts, so we cannot provide methods and results of analyses carried out on artificially infected (and non-infected) samples. In our oppion, untill now the infectious clones of two viruses have not yet been constructed in world.

Round 2
Reviewer 1 Report
Comments and Suggestions for Authors
My concerns were addressed adequately.
Comments on the Quality of English LanguageEnglish language is fine.
Author Response
Thank you for your review